# Creatine Kinase and Blood Pressure: A Systematic Review

**DOI:** 10.3390/medsci7040058

**Published:** 2019-04-09

**Authors:** L. M. Brewster, F. A. Karamat, G. A. van Montfrans

**Affiliations:** 1Creatine Kinase Foundation, POB 23639, 1100 EC Amsterdam, the Netherlands; 2Department of Vascular Medicine, Academic Medical Center, University of Amsterdam, 1105 AZ Amsterdam, the Netherlands; f.a.karamat@amc.nl; 3Department of Internal Medicine, Academic Medical Center, University of Amsterdam, 1105 AZ Amsterdam, the Netherlands; g.vanmontfrans@amc.nl

**Keywords:** creatine kinase, blood pressure, hypertension, systematic review, continental ancestry groups, antihypertensive drugs

## Abstract

**Background**: Hypertension is a main risk factor for premature death. Although blood pressure is a complex trait, we have shown that the activity of the ATP-generating enzyme creatine kinase (CK) is a significant predictor of blood pressure and of failure of antihypertensive drug therapy in the general population. In this report, we systematically review the evidence on the association between this new risk factor CK and blood pressure outcomes. **Method**: We used a narrative synthesis approach and conducted a systematic search to include studies on non-pregnant adult humans that address the association between plasma CK and blood pressure outcomes. We searched electronic databases and performed a hand search without language restriction. We extracted data in duplo. The main outcome was the association between CK and blood pressure as continuous measures. Other outcomes included the association between CK and blood pressure categories (normotension and hypertension, subdivided in treated controlled, treated uncontrolled, and untreated hypertension). **Results**: We retrieved 139 reports and included 11 papers from 10 studies assessing CK in 34,578 participants, men and women, of African, Asian, and European ancestry, aged 18 to 87 years. In 9 reports, CK was associated with blood pressure levels, hypertension (vs. normotension), and/or treatment failure. The adjusted increase in systolic blood pressure (mmHg/log CK increase) was reported between 3.3 [1.4 to 5.2] and 8.0 [3.3 to 12.7] and the odds ratio of hypertension with high vs. low CK ranged between 1.2 and 3.9. In addition, CK was a strong predictor of treatment failure in the general population, with an adjusted odds ratio of 3.7 [1.2 to 10.9]. **Discussion**: This systematic review largely confirms earlier reports that CK is associated with blood pressure and failure of antihypertensive therapy. Further work is needed to address whether this new risk factor is useful in clinical medicine.

## 1. Introduction

Hypertension is the main risk factor for cardiovascular disease, chronic kidney disease, and premature death worldwide [1]. Global prevalence of raised blood pressure in adults is estimated to be 35% to 40% [1]. The number of adults with raised blood pressure increased from 594 million in 1975 to 1.13 billion in 2015, with the increase largely in low-income and middle-income countries, rendering hypertension an important global public health challenge [1,2].

Raised blood pressure is estimated to cause 7.5 million deaths, about 13% of the total of all deaths and around 60% of cardiovascular mortality, accounting for 57 million disability adjusted life years (DALYS), or around 4% of total DALYS. Blood pressure levels have been shown to be positively and continuously related to the risk for stroke and heart disease, with the risk starting as low as 115/75 mmHg systolic/diastolic blood pressure. Complications of raised blood pressure further include heart failure, peripheral vascular disease, renal impairment, retinal hemorrhage, and visual impairment, which could largely be prevented by reducing blood pressure to 139/89 systolic/diastolic blood pressure or lower [1,3].

Blood pressure is a complex trait, affected by environmental and psychological, as well as biological factors, including education and income levels, psychosocial stress, genes, fetal and early childhood nutrition and growth, and nutrition and body mass in later life [4]. In 2000, a new genetic factor for hypertension and cardiovascular disease was proposed by our group, the activity of the ATP-generating enzyme creatine kinase (CK, EC 2.7.3.2) [5]. The enzyme transfers a phosphoryl group from creatine phosphate to ADP, thereby forming creatine and ATP, catalysing the reaction as follows [5]: 

Creatine phosphate + MgADP ↔ Creatine + MgATP

We proposed that high CK activity promotes hypertension through enhanced vascular contractility and sodium retention in the kidney [5] (Figure 1). Subsequently, CK was shown to be a significant independent predictor of blood pressure levels and the failure of antihypertensive therapy in a random sample of a multi-ethnic population of Amsterdam, the Netherlands. The subjects with the highest CK activity levels were men, persons with obesity, and persons of West-African ancestry [6,7]. Crude blood pressure increase per log CK increase was substantial at 14 mm Hg for systolic blood pressure (SBP) and 9 mm Hg for diastolic blood pressure (DBP) [6]. Since then, several other studies have reported data on this association. In this report, we systematically review the evidence on the association between this new risk factor CK and blood pressure.

## 2. Methods

We systematically reviewed the evidence on the association between CK and blood pressure in humans with standardized methods designed to conduct reviews of etiology [8]. As a consequence of the expected heterogeneity in outcomes, we used a “narrative synthesis approach”, where a narrative summary of the findings of studies is used to perform the data synthesis, with safeguards in place to avoid bias resulting from the undue emphasis on one study relative to another [9].

We used a framework with 4 elements to characterize the narrative synthesis based on Rodgers et al. [9], (a) Present the theory on the pathophysiology of the association; (b) Conduct a preliminary synthesis; (c) Explore relationships within and between studies; and (d) Critically assess the robustness of the synthesis product. Finally, we discuss the relevance of the aggregated evidence for the design of further etiological and clinical studies [8]. 

Based on the protocol presented here, we sought to identify, through systematic searching, all studies which provided original data in non-pregnant adult humans on the association between tissue or plasma CK and systemic blood pressure in population (sub) groups. We excluded case reports. We critically appraised the retrieved studies in duplo and synthesized the evidence found, including the following aspects: Population, exposure (CK), and outcome (blood pressure-related outcomes) [8]. We also analyzed confounding variables or moderators that may impact the results [8,9]. Depending on the available data, the results were planned to be synthesized in a combination of narrative and tabular summaries [8]. 

Systematic searches were conducted in February 2019, in electronic databases (Embase, PubMed, Literatura Latino-Americana y del Caribe en Ciencias de la Salud (LILACS), African Index Medicus, and IndMED), from their inception through to February 2019. Databases differed in technical search options, but a typical search strategy was, “(Creatine kinase OR Creatine phosphokinase OR HyperCKemia) AND (hypertension OR blood pressure OR Cardiovascular)”. We intended to include only studies providing original data on the association between CK and blood pressure measures and not studies merely reporting both parameters. Therefore, in developing this search strategy, we searched on title words where technically possible for a more relevant search yield, since a large majority of papers on CK and heart disease are on myocardial infarction. 

Search yields from the different databases were considered and analyzed separately to prevent merging errors and to enhance the retrieval of reports. We took further care in preventing bias in retrieval and inclusion of studies, by including reports without language restriction, and in performing a hand search. Data were extracted independently by two authors (LB and FK). Where relevant, European, African, Asian, or other descents (ancestry or ethnicity) were defined as, respectively, of European, sub-Saharan African, or Asian heritage, as indicated by the authors of the eligible papers. 

The main outcome was the association between CK as a continuous measure and blood pressure as a continuous measure, as reported by the authors. Other outcomes included the association between CK and blood pressure categories (normotension and hypertension, as defined by the authors, subdivided in treated controlled, treated uncontrolled, and untreated hypertension). 

We also assessed important determinants of plasma CK activity, such as physical exercise and the method of CK assessment. Plasma CK increases after exercise, in particular during the first 3 days, and this may dilute the association of CK with blood pressure [6]. Therefore, we collected data on whether resting CK was estimated. Furthermore, as CK estimation is a bioassay, we also collected information on whether the method of CK estimation, as reported in the papers (or from the manufacturer of the device described in the paper), was according to the standardized methods of the International Federation of Clinical Chemistry (IFCC), as previously described [10]. 

Predefined subgroups were based on sex, ancestry group, and geographical location. We expected heterogeneity in the participants’ characteristics, CK comparisons, and outcomes and therefore planned to describe the findings as reported by the authors. Data in square brackets are 95% confidence intervals and in parentheses are standard errors, unless indicated otherwise.

## 3. Results

### 3.1. Systematic Search Yield

The study flow is depicted in Figure 2 and the included studies are in Table 1 [6,7,11,12,13,14,15,16,17,18,19]. We retrieved 135 reports from electronic databases and included 7 reports from 6 studies [6,7,11,15,16,17,19]. We additionally retrieved 4 papers through hand search [12,13,14,18], 3 of which were not included in electronic databases [12,13,18]. In total, we included 11 papers from 10 studies [6,7,11,12,13,14,15,16,17,18,19].

### 3.2. Studies and Participants

The studies assessed CK in 34,578 participants, men and women of African, Asian, and European ancestry, aged between 18 and 87 years. The included studies were population studies with relatively large sample sizes (*n* = 4) [6,7,11,14,17], or clinical studies (*n* = 5) among hypertensives and controls [12,13,16,18,19]. One smaller study included participants by profession [15]. 

### 3.3. Exposure

CK was estimated under resting conditions (to control for exercise-induced hyperCKemia) in 4 out of 10 studies (2 population studies) [6,7,17,18,19]. Two additional population studies did not restrict exercise, but collected information on either the participants habitual exercise levels [11] or exercise in the 3 days before the CK test [14]. Definition of exercise varied, in “exercise or work in the past 3 days that causes large increases in breathing or heart rate if they are done for at least 10 minutes continuously” [14], “hard leisure physical exercise (sweating or out of breath) at least 2 h/week” [11], or “participants were instructed to abstain from heavy exercise during 3 days before the test. Walking, driving, and normal daily activities were allowed” [6,7,10]. In one study, subjects were asked to avoid vigorous exercise or intramuscular injection 48 hours prior to study enrollment [17]. 

CK estimation was performed with commercially available analyzers for medium- to high-volume laboratories in all but one study which did not report the method used to estimate CK [12]. Most studies (6 out of 10) used a Roche^®^ device (Table 1) [6,7,11,13,17,18,19]. Only three studies reported the application of IFCC guidelines [6,7,10,18,19]. We retrieved additional information from the supplier companies in the 7 other studies (Table 1), and all claimed traceability to the IFCC reference method [20], provided the manufacturer’s reagents are used.

### 3.4. Comparisons and Outcomes

Studies reported associations of CK with blood pressure as continuous measures, categorized blood pressure outcomes including normotensives vs. hypertensives, or the association between CK and the failure of antihypertensive treatment (Table 1). The data can be observed to be heterogeneous in participants, CK estimations, and comparisons, as expected. Therefore, we describe the data in a narrative synthesis as planned [9]. 

In 9 out of 11 papers, the direction of the outcome indicated that CK or CK MB are positively associated with blood pressure levels, hypertension (vs. normotension), or treatment failure (vs. controlled hypertension) (Table 1). Although most studies did not provide a sample size calculation, the magnitude and distribution of the outcome in relation to the sample size were sufficient to be statistically significant in the populations studied, except for 2 studies, George et al. and Mels et al., which report only subgroups under non-resting conditions (Table 1) [14,15]. 

Unlike the other included papers, the methodology of the report by George et al. was not designed to address the association of CK with blood pressure. The authors associate hypertension by sex with dichotomized CK levels at the cutoff point provided by the manufacturer, at 334 for men and 199 for women. This is remarkable, as the authors report in the same paper that these upper limits of normal (ULN) levels are incorrect. The authors report ULN for CK (IU/L) of 1001 for black men, 382 for white men, 487 for black women, and 295 for white women. The authors do not explain why they chose to dichotomize CK at 334 for men and 199 for women.

The study of Mels et al. [15] also only addressed subgroups by sex and ancestry. The mean CK (95th percentile) was respectively 127.0 (427), 115.0 (245), 75.9 (195), and 62.8 (123) IU/L for African ancestry men, European men, African ancestry women, and European women. However, at this relatively small sample size (around 100 in each group), the difference in CK by ancestry did not reach statistical significance in men and CK was only associated with blood pressure in white women (Table 1).

Notably, George and Mels et al. were the only studies that used a Beckman UniCel DxC800 (Beckman and Coulter, Germany) to estimate CK. Mels et al. additionally used the Konelab 20I Sequential Multiple Analyzer Computer (Thermo Scientific, Vantaa, Finland). Both devices have been associated with suboptimal performance, in particular in CK estimations [21,22,23,24], and, together with the non-resting conditions, this might have contributed to the reported outcome of these 2 studies, which differ in magnitude from the majority of studies included.

### 3.5. Subgroup Analysis

The association between blood pressure and CK was reported in European [6,7,11,19], Indonesian [13], Taiwanese [17], Indian [6,7,12,18], and West-African [6,7,16] populations. CK is higher in men, overweight persons, and persons of West-African-ancestry, but the studies reporting an association between CK and blood pressure outcomes provided evidence that such association is independent of sex, BMI, and ancestry, where applicable [6,7,11,17,19].

## 4. Discussion 

In this systematic review, we confirm the hypothesis [5], and subsequent finding [6] of an association between CK and blood pressure. The data from 10 studies, presented in 11 papers, were heterogeneous in characteristics of the participants, clinical conditions surrounding the estimation of plasma CK, and primary outcomes, but 9 out of 11 papers report that CK as a continuous measure is associated with blood pressure, the presence of hypertension, and/or failure of antihypertensive therapy [6,7,11,12,13,16,17,18,19]. In one study, the point estimate indicated an association between dichotomized CK and hypertension by sex, but the outcome did not reach statistical significance [14]. Another study found the association only in white women [15]. Both studies had used the Beckman UniCel DxC800 to estimate enzyme activity, which has shown less favorable results in quality assessments than the Roche device, which was used in most of the included studies. This might have impacted the magnitude and linearity of the CK estimation in these studies [20,21,22,23,24].

Normal tissue releases CK proportionate to the intracellular CK concentration, a physiologic process that occurs without tissue damage, as summarized by Brewster et al. [6]. Therefore, plasma CK in healthy persons at rest reflects tissue CK [6,7,25]. However, with exercise, lymphatic flow increases and CK from the interstitial space may enter the circulation rather abruptly, where it is cleared by the liver in around 3 days [6]. With frank tissue damage, such as after eccentric exercise, where the muscle contracts and stretches at the same time, or with myocardial infarction or brain trauma, large quantities of CK enter the circulation, proportional to intracellular CK and the damaged area [6,26,27]. Our group recently showed that these large increases in plasma CK might induce perturbations in coagulation, as circulating CK will reduce the ADP needed for platelet aggregation [27]. Hence, such high CK levels might induce coagulopathy and bleeding risk [27].

High tissue CK is thought to lead to a phenotype with greater vascular contractility and enhanced sodium retention through greater ATP buffer capacity at ATPases involved in ion transport and contractile responses [5,6,27,28,29,30]. CK is tightly bound near these ATPases, including Ca[2]^+^-ATPase, Na^+^/K^+^-ATPase, and myosin ATPase, where it rapidly provides ATP in situ. In skeletal muscle, high CK Type II muscle fibers display reduced cytoplasmic uptake of glucose and fat, which is a risk factor for glucose intolerance, insulin resistance, and obesity [31].

Thus, based on the existing evidence, the high CK phenotype might carry a greater risk for hypertension, as well as bleeding risk and obesity [27]. None of the included studies evaluated bleeding risk, and a few published studies report an association between CK and obesity [14,31,32,33], but these outcomes were not the topics of this review. CK isoenzyme distribution was reported to be normal in hypertensives [19], as expected in the absence of tissue damage.

### 4.1. Strengths and Limitations of This Study

The main strength of this narrative review is that it summarizes and discusses the existing evidence on the biologically plausible association of CK with blood pressure, indicating that after the initial report in 2006 [6], this has been found in different populations across the world.

We used rigid systematic review methodology [8,9] and took great care to retrieve and include all relevant studies, including studies not in PUBMED, studies not indexed in electronic science databases, and studies that did not find the association, without using any language restriction. We did not find evidence of bias towards a preferred outcome in the papers, the risk of which is lower with observational, non-intervention, studies where the researcher only observes certain characteristics of the sample population and records the data [8]. However, it is well known that scientific papers with “negative” findings are less likely to be published [8,9]. In particular with narrative syntheses, one needs to critically assess the presence of confounding variables or moderators that may impact on the results and we cannot exclude that publication bias affected the search yield of this review [8,9].

A further limitation of the included studies is that we are not well informed regarding the comparability and quality of the CK assays [20,21,22,23,24]. Results of blood samples assayed by routine measurement procedures should represent the true value of the sample. However, CK is not measured in moles or grams, but as catalytic and functional performance in a bioassay. The results represent a relative measure of (re-) activated enzyme activity, a method standardized by the IFCC [20]. Importantly, the level of reactivated activity and the linearity of the estimations across the spectrum of low, mid, and high CK activity may vary. This may lead to overestimation, underestimation, or distortion of linearity of the results at different levels of enzyme activity, across, but also within devices and laboratories. Therefore, laboratories need to regularly assess the quality of their CK estimation. With quality assessment, the quality of the test at low through high concentrations of standard reagents is addressed. Devices may also differ in their performance during independent quality assessments. The Roche^®^ devices scored well in quality tests [20,21,22,23,24]. Most of the devices used in the included studies were from Roche^®^, but two studies used a Beckman UniCel DxC800 device, of which the performance was reported suboptimal in independent quality assessments [21,22,23]. This might have affected the representation of the spectrum of CK values into the test results, especially the linearity at the extremes of the CK spectrum [20,21,22,23,24]. We are not well informed whether the range measured in the included studies was comparable across measuring devices and whether linearity for all assays was acceptable and comparable over the range tested [21,22,23,24]. Still, the association found between CK and blood pressure is robust and was reported across most devices used.

Furthermore, only 4 studies standardized CK assessments to resting conditions [6,7,17,18,19]. With heavy exercise, plasma CK does not well reflect tissue CK and the association with blood pressure can be expected to be attenuated under these conditions [6,30]. All studies reporting resting CK found an association with blood pressure outcomes [6,7,17,18,19]. In addition, CKMB in plasma is less dependent on exercise levels than total CK in plasma and both studies assessing the cardiac isoenzyme CKMB in plasma showed an association of CK with blood pressure outcomes [12,16]. Thus, although the molecular mechanisms of the association between CK and blood pressure outcomes are well described [5,6,27], evidence indicates that the quality of the CK estimation is relevant when analyzing the association, in taking care to test under resting conditions when using plasma CK as a surrogate for tissue CK, and in using standardized IFCC methods. The field of CK research and cardiovascular disease would benefit from the further development of non-invasive assessments of tissue CK for clinical use, such as ^31^P-magnetic resonance spectroscopy [34] of the calf muscles.

In summary, the majority of studies included in this systematic review confirm the association between CK and blood pressure. The association is based on evidence that ATP-buffer capacity is relevant for the generation of blood pressure. Relatively high CK activity may carry the risk for hypertension that is difficult to treat, and reported links of CK with obesity and bleeding risk are also biologically plausible. However, the lack of information regarding the comparability and quality of the CK assays used in the included studies is a limitation of this review. Further studies will need to address the usefulness for clinical medicine of this new and emerging field of CK related-hypertension and cardiovascular disease, which has recently been substantiated by experimental evidence providing evidence of a causal relationship [27,28,30,35], including reduction of blood pressure with CK inhibitors [35].

## Figures and Tables

**Figure 1 medsci-07-00058-f001:**
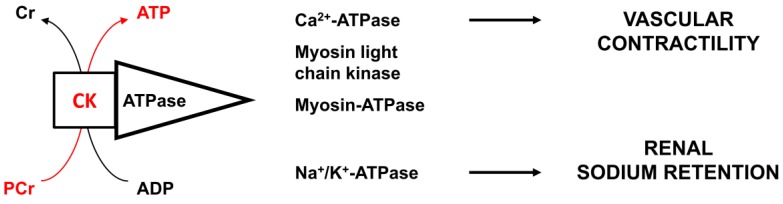
Creatine kinase and pressor responses. This figure depicts the proposed pathophysiology of high blood pressure with high creatine kinase (CK). CK is tightly bound near ATPases, such as calcium, sodium/potassium, and myosin ATPase, where the enzyme rapidly buffers the ADP generated by these ATPases into ATP, utilizing phosphocreatine. Thus, greater CK activity near these ATPases is thought to promote vascular contractility and the ability to retain sodium [5,6]. This places the individual with high CK activity at a greater risk to develop hypertension, with greater resistance against blood pressure-lowering therapy [6,7].

**Figure 2 medsci-07-00058-f002:**
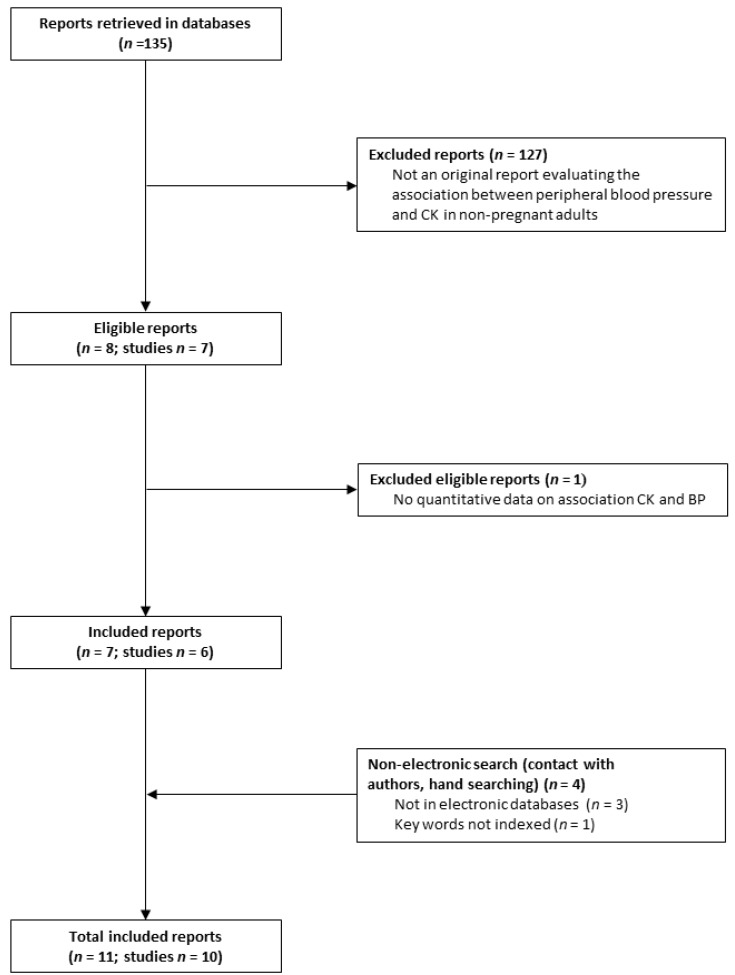
Paper Flow. The figure depicts the number of retrieved, eligible, and included reports and the yield of the hand search. The 11 included papers are reports from 10 studies.

**Table 1 medsci-07-00058-t001:** Table of Included Studies.

Author, Year	Population	Ancestry	Country	N	Age *	CK Estimations			Outcome	Effect Size ^¶^
						Resting ^†^	Device	IFCC		
**BLOOD PRESSURE**										
Brewster 2006 [6]	Random population sample	African Asian European	Netherlands	1444	35–60	Yes	Roche/Hitachi Systems	Yes ^║^	CK associated with SBP and DBP	**CK T1 (<88) vs. CK T3 (≥145)** SBP 122.5 (1.0) vs. 130.6 (0.9) DBP 79.2 (0.6) vs. 84.8 (0.6)
										**Univariable** SBP: + 13.9 [9.6 to 18.3]/log CK DBP: + 9.3 [6.8 to 11.9]/log CK
										**Multivariable** SBP + 8.0 [3.3 to 12.7]/log CK DBP + 4.7 [1.9 to 7.0]/log CK
Johnsen 2011 [11]	Population sample	European	Norway	12,776	30–87	No ^‡^	Modular P, Roche	Yes	CK associated with SBP and DBP	**CK T1 vs. CK T3** SBP 134.4 (0.4) vs. 138.2 (0.4) DBP 76.3 (0.2) vs. 79.8 (0.2)
										**Multivariable** SBP + 3.3 [1.4 to 5.2]/log CK DBP + 1.3 [0.3 to 2.3]/log CK
Mels 2016 [15]	Teachers	African	South Africa	405	45 (0.5)	No	Beckman UniCel DxC800; Konelab 20i	Yes	Only subgroup analysis	CK only associated with BP in women of European ancestry. Adjusted R2 = 0.46; β = 0.17; p = 0.03
Yen 2017 [17]	Population health survey	Asian	Taiwan	4562	49 (0.2)	Yes	Modular P, Roche	Yes	CK associated with SBP and DBP	**CK Q1 (<69) vs. CK Q4 (≥128)** SBP 118.6 (0.3) vs. 124.2 (0.3) DBP 73.1 (0.2) vs. 76.6 (0.2)
										**Univariable** SBP + 6.5 [5.2 to 7.7] CK/10 mmHg DBP + 10.1 [8.0 to 12.1] CK/10 mmHg
										**Multivariable** SBP + 1.68 CK/10 mm Hg
**HYPERTENSION**										
Brewster 2008 [19]	Cases with hyperCKemia vs. population controls	European	Netherlands	46 (controls 22,612)	18–67	Yes	Modular P, Roche	Yes ^║^	High CK associated with hypertension	**Odds ratio of hypertension ^§^** Crude: 3.9 [2.2 to 6.9] Adjusted: 2.0 [1.1 to 3.8]
Johnsen 2011 [11]	Population sample	European	Norway	12776	30–87	No‡	Modular P, Roche	Yes	CK higher with HT	CK higher in persons using anti-HT drugs vs. no anti-HT drugs (104 vs. 99)
Brewster 2013 [7]	Random population sample	African Asia European	Netherlands	1444	35–60	Yes	Roche/Hitachi Systems	Yes ^║^	CK higher in HT vs. NT	**Odds ratio of hypertension** CK T1 (<88) vs. CK T3 (≥145) HT prevalence: 26.8 vs. 41.2% Odds ratio 1.9 [1.5 to 2.5]
										**CK in HT vs. controls** CK 145.9 (7.0) HT vs. 126.8 (2.5) controls
George 2016 [14]	Population study	African Asian European	USA	10,096	>20	No	Beckman UniCel DxC800	Yes	Only subgroup analysis	**Odds ratio of HT (CK dichotomized, ULN) **** Men: 1.2 [0.8 to 1.7] Women: 1.4 [1.0 to 2.1]
Yen 2017 [17]	Population health survey	Asian	Taiwan	4562	49 (0.2)	Yes	Modular P, Roche	Yes	CK higher in HT vs. NT	**CK in HT vs. controls** CK +20.7 [15.8 to 25.6] in HT vs. controls
Sukul 2018 [18]	Hypertensives vs. controls	Asian	India	115	25–60	Yes	Roche diagnostics	Yes ^║^	CK higher in HT vs. NT	**CK in HT vs. controls** CK 199.6 (16.4) HT vs. 72.7 (4.0) controls
Sanjay Kumar 2013 [12]	Hypertensives vs. controls	Asian	India	150	40–90	No	NR	NR	CK MB higher in HT vs. NT	**CK MB in HT vs. controls** 21.5 (4.0) HT vs. 17.2 (2.4) controls
Emokpae 2017 [16]	Hypertensives vs. controls	African	Nigeria	340	28–62	No	Selectra Pro S	Yes	CK MB higher in HT vs. NT	**CK MB in HT vs. controls** 51.6 (3.0) HT vs. 15.0 (0.8) controls
**TREATMENT FAILURE**										
Johnsen 2011 [11]	Population sample	European	Norway	12776	30–87	No ^‡^	Modular P, Roche	Yes	CK not significantly higher in uncontrolled vs. controlled HT	**CK in controlled vs. uncontrolled HT** 101 vs. 110 ^††^
Brewster 2013 [7]	Random population sample	African Asian European	Netherlands	1444	35–60	Yes	Roche/Hitachi Systems	Yes^║^	CK higher in uncontrolled vs. controlled HT	**CK in controlled vs. uncontrolled HT** 124.3 (10.9) vs. 157.9 (9.4)
										**Odds ratio of treatment failure** CK T1 (<88) vs. CK T3 (≥145) HT treatment failure 46.7% vs. 72.9% Odds ratio 1.6 [1.3 to 1.9]
										**Adjusted odds ratio treatment failure** 3.7 [1.2 to 10.9]/log CK
Luman 2015 [13]	Hypertensives	Asian	Indonesia	82	>18	No	Roche/Hitachi cobas analyzer	Yes	CK higher in uncontrolled vs. controlled HT	**Mean CK in controlled vs. uncontrolled HT** 81.8 (3.3) vs. 132.2 (6.2)
										**High CK (T3 CK>109.33 U/L)** Controlled hypertension 18.5% Uncontrolled hypertension 81.5%
Sukul 2018 [18]	Hypertensives vs. controls	Asian	India	115	25–60	Yes	Roche diagnostics	Yes ^║^	CK higher in uncontrolled vs. controlled HT	**CK in controlled vs. uncontrolled HT** 99.6 (4.5) vs. 313.9 (22.5)

**Legend**. Studies reporting plasma creatine kinase (CK) and blood-pressure outcomes. Blood pressure is in mm Hg and CK in (I)U/L. Where applicable, data are rounded to one decimal place. Data in square brackets are 95% confidence intervals, in parentheses are standard errors, and outcomes are significant at *p* < 0.05, unless stated otherwise. * Age (range or mean with SE) in years, ^†^ Test under resting conditions, as defined by the authors. ‡ Outcomes adjusted for habitual exercise. IFCC, CK estimated according to the International Federation of Clinical Chemistry guidelines [20], reported by 3 studies; ^║^ [6,7,18,19] we retrieved information regarding the method of CK estimation on the internet for other studies. NR, not reported. SBP, DBP, systolic, diastolic blood pressure; HT, hypertension (as defined by the author; generally, blood pressure > 139 systolic or 89 diastolic, or the use of antihypertensive drugs). NT, normotension. CKMB, CKMB isoenzyme; ^¶^ Multivariable analyses as reported, mostly including sex, age, and BMI, among other variables; T1, T3 low vs. high CK tertile; Q1, Q4 lowest vs. highest CK quartile; **^§^** High CK compared to population controls. ** ULN, upper limit of normal (334 in men, 199 in women) [14]. †† No SE reported, *p* = 0.1, direction (one or two-sided) not reported.

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
