# Peer review of "Creatine Kinase and Blood Pressure: A Systematic Review"

_medsci, 2019, doi:10.3390/medsci7040058_

Round 1
Reviewer 1 Report
Good review and adds to the understanding of the link between CK and hypertension
Abstract
Background Hypertension is a main risk factor for premature cardiovascular death. Although blood pressure is a complex trait, we have showed that the activity of the ATP-generating enzyme creatine kinase (CK) is a signiifcant predictor of blood pressure and of failure of antihypertensive drug therapy in the general population
*edits may to avoid overstating the case for HTN-the abstract discussion is more balanced than the first few sentences that lead in
In paper I suggest change "main: to "a significant"- main indicates clear superiority of a factor as seen page 2and end of abstract- I would remove all use of "main"- overstated
Last summary paragraph is good but mention again recognizing potential confounders in CK techniques across the studies cited
Author Response
Reviewer 1. Comments and Suggestions for Authors
Comment 1. Good review and adds to the understanding of the link between CK and hypertension
Authors’ reply
We thank the reviewer for the time spent to improve the manuscript.
Comment 2. Abstract
Background Hypertension is a main risk factor for premature cardiovascular death. Although blood pressure is a complex trait, we have showed that the activity of the ATP-generating enzyme creatine kinase (CK) is a significant predictor of blood pressure and of failure of antihypertensive drug therapy in the general population
*edits may to avoid overstating the case for HTN-the abstract discussion is more balanced than the first few sentences that lead in
In paper I suggest change "main: to "a significant"- main indicates clear superiority of a factor as seen page 2and end of abstract- I would remove all use of "main"- overstated
Authors’ reply
We removed the word “main” in association with CK and BP throughout the paper and used “significant” and other wording (marked).
Comment 2. Summary Paragraph
Last summary paragraph is good but mention again recognizing potential confounders in CK techniques across the studies cited
Authors’ reply
We thank the reviewer for this suggestion. We added “However, the lack of information regarding the comparability and quality of the CK essays used in the included studies is a limitation of this review” to the summary paragraph (Line 143, 144; marked).
Reviewer 2 Report
This is a very nice overview on the relationship between CK and blood pressure/hypertension. We must realize, however, that the number of papers on which the present review is based, is still low. This calls for some caution. In particular, one would like to know to what extent publication bias may have occurred. Although the authors mention that they did not find evidence for a 'preferred outcome' in the papers, they should elaborate a bit more on this potential confounder.
Author Response
Reviewer 2. Comments and Suggestions for Authors
Comment. This is a very nice overview on the relationship between CK and blood pressure/hypertension. We must realize, however, that the number of papers on which the present review is based, is still low. This calls for some caution. In particular, one would like to know to what extent publication bias may have occurred. Although the authors mention that they did not find evidence for a 'preferred outcome' in the papers, they should elaborate a bit more on this potential confounder.
Authors’ reply
We thank the reviewer for this constructive comment. Indeed we cannot exclude publication bias and we will acknowledge this more explicitly in the paper. Two out of 11 reports included did not show the association, and these were published as well.
‘But indeed, there is certainly bias in that authors are less likely to submit negative findings and journals are less likely to accept and publish these papers. We edited the limitation section to make this clearer to the reader (Line 104 to 108, marked).
Authors’changes
Small textual edits were made throughout the paper.